# Understanding the Uncivilized Tourism Behavior of Tourists: A Planned Behavior Model Based on the Perspectives of Cognitive Dissonance and Neutralization

**Ping Zhang [1] and Kaijun Cao [2,3,***

1    School of Business, Xinjiang University, Urumqi 830091, China
2    School of Tourism Studies, Xinjiang University, Urumqi 830046, China
3    Key Laboratory of Sustainable Development of Xinjiang's Historical and Cultural Tourism, Xinjiang University, Urumqi 830046, China
*    Correspondence: caokaijun@xju.edu.cn

**Abstract:** Effectively regulating and managing the uncivilized tourism behavior of tourists is a key strategy for the sustainable development of tourism destinations. In this paper, the dissonance–neutralization model was proposed by integrating planned behavior, cognitive dissonance, and neutralization theories. Partial least squares structural equation modeling (PLS-SEM) was used to test the impact and effect of this model on the uncivilized tourism behavior of 387 tourists at natural heritage sites. Overall, the research results show the following: (1) Uncivilized tourism behavior is not only determined by behavioral intention and perceived behavioral control. Attitude, subjective norms, and perceived behavioral control also all have a significant influence on behavioral intention. (2) Cognitive dissonance is a parallel predictor of behavior. (3) Neutralization techniques can effectively reduce cognitive dissonance, thus allowing uncivilized tourist behavior to continue.

**Keywords:** uncivilized tourism behavior of tourists; theory of planned behavior; neutralization techniques; cognitive dissonance; natural heritage sites

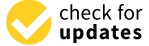



## 1. Introduction

The uncivilized tourism behavior of tourists is an important problem facing the tourism industry. Uncivilized tourism behavior means that tourists fail to abide by social norms and engage in negative activities in the process of traveling [1], damaging the resources and humanistic environment of tourism destinations [2,3]. Uncivilized tourism behavior is fairly common in a variety of countries and regions, including such examples as walking outside trails [4], damaging local cultural traditions and customs [1], throwing litter about [5], destroying cultural relics and historic sites [6], scrawling [7], making noises, disturbing others [8], etc. These behaviors, to a certain extent, lead to the deterioration of the environment, reduce the recreational quality of fellow tourists, and increase the management cost of the recreation environment [9,10]. Johnson and Kamp advised that the United States National Parks spend USD 79 million a year on repair and USD 18 million on regular cleaning and maintenance due to the uncivilized tourism behavior of tourists [11]. Still, having said this, only a few scholars have paid enough attention to this phenomenon [1,2]; furthermore, only certain scholars have attempted to investigate the uncivilized tourism behavior of tourists. However, most of their studies on this topic remain descriptive [12] and tend to adopt qualitative and observational methods instead of quantifying non-guiding important determinants [4]. The existing literature mainly involves the concept definition [1], negative impact [13], causes of existence [2,14], prevention strategies, and other aspects of uncivilized tourism behavior. There has been little research that has focused on the decision-making process behind the uncivilized tourist behavior

of tourists. In addition, only a few complete theoretical or empirical studies have been conducted on the causes of uncivilized tourism behavior [4,15,16].

The uncivilized tourism behavior of tourists is a decision that is made in a violation of the norms. The decision-making process for this type of behavior includes both will factors and incomplete will factors [4]. For instance, a relaxed attitude toward authoritative rules, such as toward laws and customs, stems from the influence of family and society, as well as the perceived difficulty of constraints and control because of external circumstances (e.g., the lack of resources). These factors can be classified as the research framework for the theory of planned behavior (TPB) [17]. The theory of planned behavior allows an exploration of the formation process of individual behavior by taking into account both will factors and incomplete will factors, which are, in turn, widely used in the field of tourism research [18,19]. Despite helping to better understand uncivilized tourism behavior, the theory of planned behavior fails to explain the reasons for the uncivilized tourism behavior of tourists [2], as certain tourists may have a sense of dissonance when doing something that is against the norms. However, the emotional response of cognitive dissonance may induce uncivilized tourism behavior.

According to the theory of cognitive dissonance, individuals usually adjust their cognition to relieve uncomfortable psychological states and to change their attitude or behavior when cognitive factors are contradictory [20]. As mentioned above, tourists may engage in uncivilized tourism behavior after balancing these mutually exclusive cognitive states (e.g., the constraints of moral norms and the convenience of deviating from sightseeing trails) and these cognitive factors. Therefore, uncivilized tourism behavior may be ascribed to the cognitive dissonance of tourists. Nevertheless, though cognitive dissonance may account for the contradiction between the cognition and behavior of tourists, the reason why tourists can break away from cognitive dissonance remains unknown—tourists still need to find reasonable excuses for their violations of the norms.

The neutralization technique theory explains the process in which individuals use rational justification [21]. This theory holds that individuals will master certain skills to offset behavioral nature and to rid themselves of the constraints of social norms in order to successfully perform activities that violate norms but which are not criminal [22]. Thus, tourists may rationalize their uncivilized tourism behavior without realizing their mistakes. Furthermore, they may even deny the consequences they have caused. This phenomenon is closely related to the mechanism of cognitive dissonance reduction [15]. The emotional response of cognitive dissonance may occur when tourists behave in such a way as to perform a violation of social norms. To avoid cognitive dissonance, tourists will find reasons to support the implementation of uncivilized tourism behavior to alleviate their sense of dissonance. Zhang et al. found that neutralization techniques reduce the cognitive dissonance of tourists that result from the violation of norms [15]. Neutralization techniques are a way for tourists to wriggle out of cognitive dissonance. Hence, cognitive dissonance and neutralization techniques are important determinants for tourists' uncivilized tourism behavior.

This study was aimed at exploring the impact and effects of the rational TPB framework, as well as the theories of cognitive dissonance and neutralization techniques on the uncivilized tourism behavior of tourists. First, the theory of planned behavior was taken as the basic analysis framework. Then, attitude, subjective norms, perceived behavioral control, and behavioral intention were taken as the important influence factors for the uncivilized tourism behavior of tourists. Second, the initial TPB model was expanded, and the theories of cognitive dissonance and neutralization techniques were added as predictive factors to the TPB model. Finally, the planned behavior model was constructed from the perspectives of cognitive dissonance and neutralization. As such, the impact and effects of different factors on the uncivilized tourism behavior of tourists were verified. This study was conducted at a natural heritage site in Xinjiang, China. The research results of this study will help scholars and practitioners to better understand the uncivilized tourism behavior of tourists.

## 2. Literature Review

### 2.1. TPB

TPB is a derivative theory of the theory of reasoned action (TRA) and is one of the most influential theoretical models that explains the decision-making process of individuals [17]. TRA hinges only on the subjective will of individuals. However, most behaviors are not only determined by the subjective willingness of individuals, but also depend on the behavioral control of individuals, such as time, money, and skills [23]. As a result, Ajzen [23] introduced the concept of perceived behavioral control in order to construct TPB. The theory involves a relatively deliberate process of considering individual costs and benefits from participating in various behaviors in order to predict individual behaviors that are free from the control of the completely subjective will.

TPB has been extensively used to explain behavior in tourism. This kind of behavior includes the willingness to revisit tourism destinations [24], tourists' travel mode choice [25], green purchase behavior [26], accommodation purchase decisions [27], the intention of participating in low-carbon tourism [28], etc. Furthermore, certain studies have improved the predictive power of the model by adding variables in addition to attitude, subjective norms, and perceived behavioral control. For example, tourism motivation, reputation, image, etc., were added to expand the TPB model and to confirm the enhancement of its predictive power [24]. Panwanitdumrong et al. noted that the environmental awareness and background of tourists will increase the effectiveness of the model in terms of environmentally responsible behavior [29]. Similarly, more than 5% of the difference in behavioral intention was explained in the analysis of tourists' motivation to travel [30]. Therefore, this study took TPB as a reference point for the uncivilized tourism behavior of tourists and proposes to add these additional factors to its structure and its interaction.

### 2.2. Theory of Cognitive Dissonance

The description of the theory of cognitive dissonance, as proposed by Festinger, is based on daily life: An individual feels psychologically uncomfortable and is usually inclined to reduce the sense of dissonance when having two or more inconsistent cognitive states. Mutually exclusive cognitive states may result in the production of negative emotional responses when tourists decide to engage in uncivilized tourism behavior. Cognitive dissonance is often reduced [20] and opinions are adjusted to reduce inconsistent factors in order to avoid cognitive dissonance. Neutralization techniques are even adopted to avoid increasing the causes of dissonance in order to alleviate cognitive dissonance.

Sweeney et al. noted the existence of cognitive and emotional dissonance [31]. The cognitive component refers to the choice of individuals between attractive and repellent items, as well as the occurrence of post-decision dissonance [31]. Meanwhile, the emotional component represents a state of psychological discomfort [32]. Tourists will have psychological discomfort and seek the mechanism of dissonance reduction when their cognitive factors diverge. This is strongly linked with neutralization techniques that are strategies for reducing cognitive dissonance [33]. As suggested by Hinduja, consumers can recognize that software piracy is a consumption behavior that violates norms and is opposite to the values accepted in their hearts, but still find excuses to justify their behavior and to counteract the constraints of social norms [34]. Likewise, Zhang et al. found that conflicts between environmental and driving regulations usually give rise to the cognitive dissonance of tourists whose psychological discomfort is reduced with the help of neutralization techniques [15]. In addition, qualitative research by Uba and Chatzidakis discovered that college students will use neutralization techniques to ease dissonance, thereby using the techniques to prove that driving is less harmful to the environment [22]. Tourists who violate tourism regulations may use neutralization techniques to "rationalize" uncivilized tourism behavior. In this study, therefore, neutralization techniques were connected with cognitive dissonance as a way for tourists to reduce cognitive dissonance.

*2.3. Neutralization Technique Theory*

Originating from social psychology, the neutralization technique theory was first proposed by Sykes and Matza [21]. They utilized this theory to explain, in essence, deviant behavior that is conducted in violation of social norms and constraints. Individuals may use certain language skills to deflect blame for bad behavior and relieve stress when violating social norms. These "neutralization techniques" also allow continued violations to be possible. Neutralization techniques have seen extensive applications in a variety of fields, such as consumption misbehavior [35], employees' violation of norms [36], dangerous sports, and tourists' pro-environmental behavior [15,37]. The qualitative research of Uba and Chatzidakis showed that neutralization techniques can also explain the environmental destruction behavior of college students [22]. Park et al. studied employees' violation of work norms and found the widespread use of neutralization techniques in their defense [38]. In a similar manner, Zhang et al. noticed that neutralization techniques can predict the pro-environmental behavior of tourists [15]. Five neutralization techniques were proposed by Sykes and Matza, which have been further supported by current studies. These techniques mainly include the denial of responsibilities, the denial of injuries, the denial of victims, the condemnation of condemners, and the appeal to higher loyalties [21]. Subsequent research has expanded the application scope of neutralization techniques and added five additional techniques: the claim of normalcy, relative acceptability, relative entitlement, the defense of necessity, and the change locus of control argument [22].

In the case of uncivilized tourism, the behavior of tourists violates social norms (these norms are also usually binding on the tourists). Neutralization techniques may be used by tourists to justify their uncivilized tourism behavior. Based on the research of Uba and Chatzidakis, as well as Park et al. [22,38], neutralization technique dimensions can be conceptualized as a formative second-order structure. Three or four dimensions are sufficient to define neutralization techniques. Multi-dimensional structures allow researchers to make a choice according to different research fields and to express neutralization techniques in multiple sub-dimensions. Thus, this study combined the actual situation of the research site and selected four techniques that may be used by tourists to prove their engagement in uncivilized tourism behavior. Among these techniques, the denial of responsibility means that tourists refuse to take responsibility for their misbehavior and believe that other factors lead to the implementation of their misbehavior [21]. The denial of injury means that tourists claim that misbehavior is harmless and the victims (i.e., scenic spots) can fully afford the damage caused by the above behavior [21]. Tourists perceive that the appeal to higher loyalties aims to achieve more important interests and can ignore and undermine the demand of the whole society. Thus, they use the change locus of control argument to illustrate that no substantial changes will be made to scenic spots even if they stop misbehavior [22].

## 3. Research Hypotheses and Model Building

In various fields, a majority of studies have used TPB to predict individual behavior. The attitude toward, subjective norms, and the perceived behavioral controls of behavior are determinants for behavioral intention in the decision-making process of individuals [39,40]. Researchers have confirmed the importance of these variables in predicting the behavioral intentions of tourists [41], but the behavior itself was rarely measured. From the perspective of Ajzen, the stronger intention individuals have for a behavior, the more likely they will be to implement the behavior [23]; perceived behavioral control will beget stronger behavioral intention and behavior [23]. Previous studies have indicated that the perceived behavioral control of tourists can predict their civilized tourism behavioral intentions and behaviors [23,42], as behavioral intention plays a mediating role between subjective norms and behavior [43]. According to the literature review, the following hypotheses are proposed:

**H1:** *Attitude has a positive impact on behavioral intention;*

**H2:** *Subjective norms have a positive impact on behavioral intention;*

**H3:** *Perceived behavioral control has a positive impact on behavioral intention;*

**H4:** *Behavioral intention has a positive impact on uncivilized tourism behavior;*

**H5:** *Perceived behavioral control has a positive impact on uncivilized tourism behavior;*

**H6:** *Behavioral intention plays a mediating role in the impact of perceived behavioral control on uncivilized tourism behavior.*

Subjective norms are the external pressure felt by tourists when they engage in uncivilized tourism behavior. Such external pressure can promote and restrain the implementation of their behavior [44]. However, the uncivilized tourism behavior of tourists is bound to bring a negative image impact and is unlikely to be regarded as a "virtue" without negative qualities. For this reason, uncivilized tourism behavior experiences at least some cognitive dissonance that is brought about by external pressure. Thus, the following hypothesis was proposed:

**H7:** *Cognitive dissonance plays a mediating role in the impact of subjective norms on uncivilized tourism behavior.*

The occurrence of psychological discomfort will impel individuals to use the strategies of cognitive dissonance reduction, with a view to adjusting the gap between uncivilized tourism behavior and their own self-image [15]. The use of neutralization techniques includes not only techniques such as "appeal to higher loyalties" but also skills reducing inconsistent factors such as "normal behavior". Hence, neutralization techniques are an option for addressing cognitive dissonance. As such, the following hypothesis was proposed:

**H8:** *Neutralization techniques have a negative impact on cognitive dissonance.*

In the early research of the free choice paradigm (FCP), another effect of cognitive dissonance is in being associated with attitude, from which the existence of dissonance is often derived. Therefore, the expectation regarding the degree of behavior-related dissonance will have a bearing on the attitude toward behavior [45], which is consistent with the theory of cognitive dissonance. The higher the degree of dissonance is, then the stronger the pressure to engage in more uncivilized tourism behavior will be. During cognitive dissonance reduction, individuals have tried neutralization techniques, as well as other methods. If cognitive dissonance persists, the choice to give up positive attitudes will be performed in order to restore cognitive consistency. Thus, the following hypotheses were proposed:

**H9:** *Cognitive dissonance has a negative impact on attitude;*

**H10:** *Cognitive dissonance plays a mediating role in the impact of neutralization techniques on attitude.*

As mentioned earlier, cognitive dissonance is a state of holding conflicting perceptions. With the aid of neutralization techniques, tourists can identify themselves as people with anomie and engage in uncivilized tourism behavior in the future. The arousal of dissonance does not mean performing the behavior, but merely recognizing that it conflicts with social norms. Tourists without the idea of engaging in uncivilized tourism behavior may be trapped in an uncomfortable state, such as being enticed into thinking about this problem. Tourists who do not intend to perform the behavior may develop cognitive dissonance according to their opinion of the behavior [20]. Therefore, cognitive dissonance was deemed to be the antecedent of uncivilized tourism behavior, after these factors were taken into consideration. As such, the following hypothesis was proposed:

**H11:** *Cognitive dissonance has a negative impact on uncivilized tourism behavior.*

To sum up, TPB is a rational decision-making theory that is commonly used to analyze tourist behavior. In accordance with TPB, individual behavior is controlled by behavioral

intention, and behavioral intention is influenced by the three variables of attitude, subjective norms, and perceived behavioral control; further, behavioral intention and behavior come from perceived behavioral control. The TPB mainly includes attitude, subjective norms, behavioral intention, and behavior. The decision-making model of tourists' uncivilized tourism behavior is shown in Figure 1.

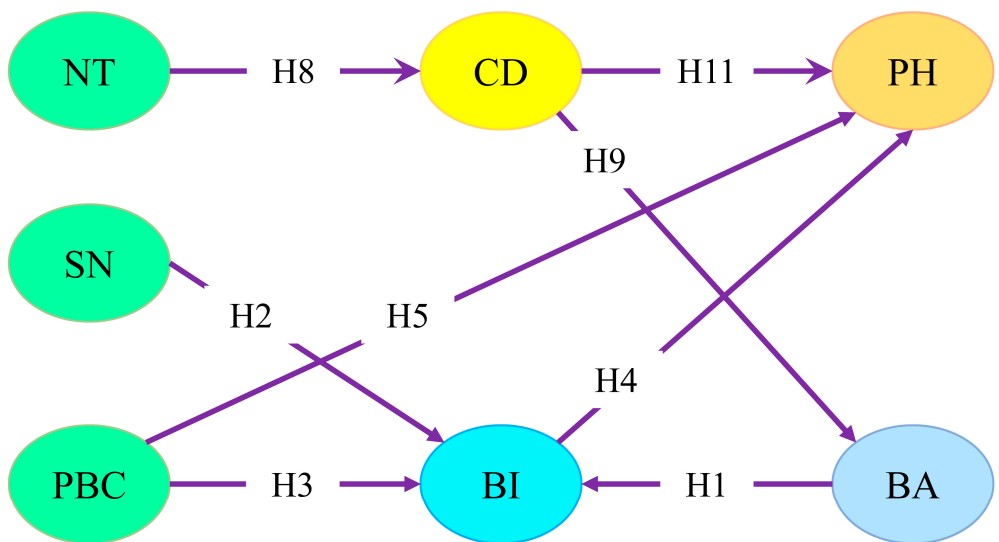

**Figure 1.** Research Model. Note 1: H6, H7, and H10, which are on mediating effect, are not indicated in the figure; Note 2: Abbreviations—BA: attitude toward uncivilized tourism behavior; SN: subjective norms of uncivilized tourism behavior; PBC: perceived control of uncivilized tourism behavior; BI: intention of uncivilized tourism behavior; PH: uncivilized tourism behavior; NT: neutralization techniques; and CD: cognitive dissonance.

The attitude toward uncivilized tourism behavior derives from the beliefs regarding attitude objects. Beliefs are related to the properties of objects, thereby associating behavior with certain consequences (e.g., the convenience of crossing trails at will). Favorable attitudes will be formed if tourists believe that uncivilized tourism behavior can produce desired results.

Tourists are also influenced by family or social attributes and are constrained by laws, moral concepts, and the management norms of tourism destinations. As a result, the relevance of tourists to family or society may affect the intensity of subjective norms. Perceived behavioral control has something to do with the resource management and control ability of tourists' uncivilized tourism behavior. This part is also in connection with the difficult degree of the behavior performed in addition to the resource status of tourists. Additionally, the behavioral intentions of tourists are the predictive indicators that are closest to behavior.

The relationships between attitude, cognitive dissonance, and behavior in the context of uncivilized tourism were built according to the theory of cognitive dissonance that was proposed by Festinger. The emotional response of cognitive dissonance will occur when the attitudes of tourists conflict with social norms. Tourists may attempt to change their attitudes or behavior to reduce cognitive dissonance. In TPB, the attitudes of tourists have an influence on behavioral intention. In this study, the neutralization technique theory, as raised by Sykes and Matza, was also applied to construct the relationship between neutralization techniques and cognitive dissonance. Various kinds of neutralization techniques may be strategies that are used to reduce cognitive dissonance when tourists violate social norms. Prior studies have proved, in different contexts, that neutralization techniques can counteract the effect of cognitive dissonance. Thus, the theories of cognitive dissonance and neutralization techniques should be taken into account to better understand the uncivilized tourism behavior of tourists.

The original TPB had limited research on this behavior. It is important that the TPB model allows an expansion of the boundary of rational research by adding other theories [23], as shown in Table 1. Research on individual behavior showed that tourists are more likely to engage in unconstrained behavior when visiting natural heritage sites [46]. The theories of cognitive dissonance and neutralization techniques in psychological theories may be more important than other theories. In view of this, this study expanded the initial TPB model and considered the situation of uncivilized tourism behavior and the characteristics of tourism destinations. The theories of cognitive dissonance and neutralization techniques were combined with the framework of rational behavior to predict the uncivilized tourism behavior of tourists.

**Table 1.** Impact of TPB, NT, and CD on BI in tourism research.

| Author(s) | Research Scenarios | TPB | CD | NT | Other Variables |
|---|---|---|---|---|---|
| Wang et al. [47] | Visiting green hotels | BA→BI ** SN→BI *** PBC→BI * | | | Perceived effectiveness and environmental concerns |
| Han [48] | Staying at a green hotel | BA→BI ** SN→BI ** PBC→BI ** | | | TPB and value-belief-norm theory |
| Panwanitdumrong, Chen [29] | Environmentally responsible behavior | BA→BI ** SN→BI *** PBC→BI *** PBC→BI→PH *** | | | Environmental awareness and environmental backgrounds |
| Wang et al. [44] | Environmentally responsible behavior | BA→BI * SN→BI * PBC→BI * PBC→BI→PH * | | | Environmental interpretations |
| Zhang et al. [15] | Pro-environmental behavior | BA→BI *** SN→BI * | | NT→BI *** | Perceived benefits, negative perceptions |
| Fang et al. [45] | Purchase behavior | | CD→BI *** | | Negative moods |

Note: * $p < 0.05$, ** $p < 0.01$, and *** $p < 0.001$.

## 4. Research Design

### 4.1. Questionnaire Design

The research questionnaire consists of two parts. The first part includes the measurement items of each latent variable, which were derived from previous studies and then back-translated into English entries. In addition, tourism experts (two university teachers, who were engaged in teaching and research in the study of tourism, and one scenic spot administrator) were invited to re-discuss ambiguous statements in order to improve accuracy. The measurement of tourists' attitudes, subjective norms, perceived behavioral controls, and behavioral intentions referred to the research of Ajzen, Han et al., and Uba et al. [22,23,48]. The measurement of neutralization techniques was conducted in reference to the research of Sykes et al. [21] and Uba et al. [22]. Zhang et al. investigated the specific neutralization techniques used by tourists in the context of tourism [15]. Siponen and Vance opined that the dimensions of each neutralization technique are interchangeable, each of which represents different aspects of neutralization techniques [49]. The measurement items of the neutralization techniques should be determined according to the differences between the different scenic spots [15]. Based on the application experience of the neutralization theory in different fields, this study is combined with the characteristics of the Xinjiang Tianshan Tianchi World Heritage Site (hereafter referred to as "Tianshan Tianchi Scenic Spot). In this study, opinions were solicited from experts. The denials of responsibility and injury, the appeal to higher loyalties, and the change locus of control argument were selected as the measurement parameters of the neutralization techniques. The measurement of cognitive dissonance came from the study of Lee [50], while that of the tourists' uncivilized tourism behavior was mainly understood as per the study of Chang

et al. [51]. All measurement items adopted a five-point Likert scale. The second part of the measurement is in the basic statistical information of the respondents, including gender, age, education level, etc.

### 4.2. Data Collection

The Tianshan Tianchi Scenic Spot was chosen as the case site in this study. Based on natural ecosystem resources such as forests, glaciers, and lakes, the Tianshan Tianchi Scenic Spot has a complete vertical natural landscape with four distinctive seasons. It has been listed as a man-and-biosphere reserve, as well as a world natural heritage site by the United Nations Educational, Scientific and Cultural Organization. At present, the Tianshan Tianchi Scenic Spot has become a must-visit place for the numerous Chinese tourists to Xinjiang [52], as shown in Figure 2. However, the uncivilized tourism behavior of tourists has already touched the local ecological protection red line. On account of this, a survey was conducted on tourists who visited the Tianshan Tianchi Scenic Spot last year in order to understand the uncivilized tourism behavior of tourists in the natural heritage site.

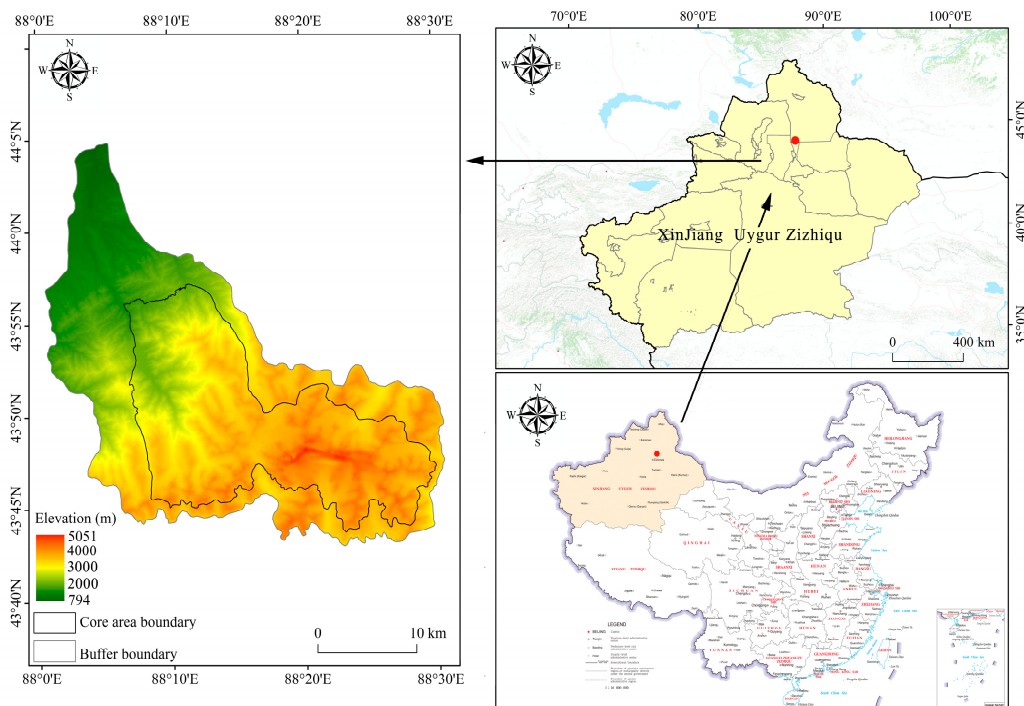

**Figure 2.** General Situation and Geographical Location of the Tianshan Tianchi Scenic Spot. Note: The red dot represents its geographical location in China.

Before the formal questionnaire was issued, the initial scale was adjusted and optimized by the feedback from three people visiting the Tianshan Tianchi Scenic Spot. Then, the scale was sent via the questionnaire link to the Alipay intangible cultural heritage guardian group, which is formed of tourism groups the respondents and family members, classmates, etc. have participated in. This was conducted and met response requirements on 25 April 2022. A total of 130 valid pre-survey questionnaires were collected, and 112 valid questionnaires remained after the exclusion of incomplete questionnaires. The criteria for deleting items were as follows: (1) the evaluation sample reliability assessed by Cronbach's alpha is less than 0.7; (2) each factor loading is less than 0.5; or (3) the double factor loading is greater than 0.4. Ultimately, a formal scale with 33 items was formed.

Formal data were collected mainly through the Wenjuanxing platform. The link to the questionnaire was posted to the Baidu post bar of Tianshan Tianchi and the section on travel tips on websites such as Qunar. A total of 468 questionnaires were sent out, and 387 valid questionnaires were finally retrieved, with an effective rate of 82.6%.

### 4.3. Sample Description

Among 387 valid samples, the number of males was greater than that of females (53.7% males and 46.3% females). One possible reason for this phenomenon is that web link surveys generally have a relatively low response rate, thereby leading to a higher proportion of male respondents than female ones. A majority of tourists were aged between 26 and 40 (43.9%) and held bachelor's degrees (31.5%). In terms of occupational composition, students (24.5%) accounted for the highest proportion, followed by freelancers (16%). Most tourists had a monthly personal income of CNY 4001 to 6000 (34.6%). In regard to the demographic characteristics of tourists, most of them came from provinces other than Xinjiang (83.4%) and had traveled to the Tianshan Tianchi Scenic Spot for the first time (81.9%). As illustrated in Table 2, the sample information objectively reflected the basic tourism profile of the Tianshan Tianchi Scenic Spot.

**Table 2.** Basic Tourism Profile of the Tianshan Tianchi Scenic Spot.

| Features | Classification | Frequency | Percentage |
| --- | --- | --- | --- |
| Gender | Male | 208 | 53.7% |
| | Female | 179 | 46.3% |
| Age | Below 18 | 2 | 0.5% |
| | 18–25 | 82 | 21.2% |
| | 26–40 | 170 | 43.9% |
| | 41–60 | 71 | 18.3% |
| | 61 and above | 62 | 16% |
| Education level | Junior high school and below | 37 | 9.6% |
| | High school | 64 | 16.5% |
| | College | 72 | 18.6% |
| | Undergraduate | 122 | 31.5% |
| | Postgraduate and above | 92 | 23.8% |
| Occupation | Corporate Commercial staff | 33 | 8.5% |
| | Officials | 23 | 5.9% |
| | Technicians | 60 | 15.5% |
| | Teachers | 26 | 6.7% |
| | Freelancers | 62 | 16% |
| | Students | 95 | 24.5% |
| | Retirees | 61 | 15.8% |
| | Unemployed | 1 | 0.3% |
| | Other | 26 | 6.7% |
| Monthly income | Below 2000 | 75 | 19.4% |
| | 2001–4000 | 103 | 26.6% |
| | 4001–6000 | 134 | 34.6% |
| | 6001–8000 | 34 | 8.8% |
| | Above 8000 | 41 | 10.6% |
| Number of visits to the Tianshan Tianchi Scenic Area | Once | 317 | 81.9% |
| | Twice | 50 | 12.9% |
| | Three or more times | 20 | 5.2% |

## 5. Data Analysis

### 5.1. Testing of Measurement Models

The normal distribution of data for scale reliability and validity was tested by the use of SmartPLS 3.0 statistical software and was based on partial least squares structural equation modeling (PLS-SEM). As was advised by Wells, Taheri, and Gregory-Smith [53], PLS-SEM was used in the following situations: (1) where the structural models are complex; (2) where the structural models contain reflective and formative indicators; and (3) where the data

conforms to a non-normal distribution. In the present study, neutralization techniques were conceptualized as the second-order formative indicators of first-order reflective indicators. The research model was complex with seven latent variables and 33 indicators. The data of one indicator followed a non-normal distribution. Hence, SmartPLS software was used.

PLS-SEM provides non-parametric evaluation criteria. According to the suggestion of Hair et al. [54], all first-order variables were measured in this study. The results are shown in Table 3. The reliability and validity of first-order variables were evaluated. The composite reliability (CR) of the measurement model was found to be greater than 0.7, thus indicating good internal consistency. The average variance extracted (AVE) was greater than 0.5 and the factor loading of the measurement items was greater than 0.7, thereby indicating that convergent validity was met.

**Table 3.** Test Results of the Measurement Model.

| Latent Variables and Measurement Metrics | Mean | Skewness | Kurtosis | Loading | t-Value | CR | AVE |
|---|---|---|---|---|---|---|---|
| **Attitudes toward uncivilized tourism behavior (BA)** | | | | | | 0.938 | 0.835 |
| Off-route travel in scenic areas is beneficial; | 3.773 | −0.087 | −0.734 | 0.918 | 92.373 | | |
| It is wise to travel off the beaten track in scenic areas; | 3.69 | −0.148 | −0.591 | 0.904 | 80.062 | | |
| Off-route travel in scenic areas is satisfying. | 3.817 | 0.342 | −0.929 | 0.92 | 99.56 | | |
| **The subjective norm of uncivilized tourism behavior (SN)** | | | | | | 0.937 | 0.832 |
| Many people around me travel off the beaten track in scenic areas; | 3.597 | −0.495 | −0.414 | 0.908 | 109.224 | | |
| It is common for people around me to travel off the beaten track in scenic areas; | 3.496 | −0.471 | −0.253 | 0.906 | 99.546 | | |
| Many people around me think that off-route travel is a good idea. | 3.488 | −0.309 | −0.385 | 0.922 | 113.68 | | |
| **Perceived control of uncivilized travel behavior (PBC)** | | | | | | 0.921 | 0.796 |
| I can easily travel off the beaten track in scenic areas if wanting to; | 3.261 | −0.07 | 0.277 | 0.871 | 55.984 | | |
| I think taking an off-route trip can be entirely up to me; | 3.55 | −1.078 | −0.191 | 0.909 | 88.935 | | |
| I feel that I have the knowledge, skills, and other resources needed to travel off-route in scenic areas. | 3.367 | −0.428 | −0.005 | 0.896 | 70.816 | | |
| **Intention of uncivilized travel behavior (BI)** | | | | | | 0.911 | 0.772 |
| I plan to travel off-route on subsequent trips; | 3.401 | −0.343 | −0.278 | 0.888 | 80.073 | | |
| On the rest of the trip, I may travel off the beaten track; | 3.442 | −0.488 | −0.241 | 0.883 | 78.045 | | |
| I am willing to travel off the beaten track for the rest of the trip. | 3.442 | −0.221 | −0.242 | 0.866 | 57.898 | | |
| Latent Variables and Measurement Metrics | Mean | Skewness | Kurtosis | Loading | t-Value | CR | AVE |
| **Uncivilized travel behavior (PH)** | | | | | | 0.928 | 0.719 |
| During the tour, I did not follow the designated route and entered an unexplored area; | 3.791 | −0.125 | −0.811 | 0.854 | 49.131 | | |
| I will make use of tourist facilities and bring equipment for myself when traveling; | 3.669 | −0.648 | −0.508 | 0.854 | 45.717 | | |
| I will trample the lawn during the tour; | 3.607 | −0.467 | −0.351 | 0.836 | 48.018 | | |
| I call the waiter loudly and freely when dining on tour; | 3.687 | −0.497 | −0.541 | 0.852 | 52.951 | | |
| I stay in my seat for a long time without leaving when finishing a tour meal. | 3.667 | −0.421 | −0.5 | 0.845 | 48.502 | | |

**Table 3.** *Cont.*

| | | | | | | | |
|---|---|---|---|---|---|---|---|
| **Cognitive dissonance (CD)** | | | | | | 0.916 | 0.731 |
| I would feel depressed; | 2.37 | −0.398 | 0.269 | 0.85 | 57.394 | | |
| I would feel I was wrong; | 2.111 | −0.168 | 0.726 | 0.857 | 58.047 | | |
| Comparing off-route travel with what I expected, I would feel like I made a bit of a bad decision on this trip; | 2.302 | −0.276 | 0.369 | 0.879 | 67.065 | | |
| I do not think an off-the-beaten-track tour matches what others have described. | 2.292 | −0.324 | 0.36 | 0.834 | 43.48 | | |
| **Denial of responsibility (DOR)** | | | | | | 0.925 | 0.804 |
| Off-route tours due to the lack of public facilities in scenic areas; | 3.553 | −0.693 | −0.306 | 0.905 | 100.928 | | |
| Off-route travel due to poor public services in scenic areas; | 3.599 | −0.483 | −0.397 | 0.909 | 105.51 | | |
| Off-route travel due to the lack of information on public services in scenic areas. | 3.669 | −0.408 | −0.465 | 0.875 | 70.324 | | |
| **Denial of injury (DOI)** | | | | | | 0.928 | 0.811 |
| My actions cause no damage to the environment; | 3.576 | −0.512 | −0.375 | 0.912 | 118.119 | | |
| My actions cause very little damage to the environment; | 3.566 | −0.349 | −0.34 | 0.9 | 90.793 | | |
| My actions cause negligible damage to the environment. | 3.545 | −0.397 | −0.367 | 0.889 | 81.844 | | |
| **Appeal to higher loyalties (AHL)** | | | | | | 0.935 | 0.826 |
| My family likes to travel off the beaten track in scenic areas to take nice photos; | 3.612 | −0.935 | −0.407 | 0.918 | 122.114 | | |
| My friend likes to travel off-route in scenic areas to avoid crowds; | 3.672 | −0.348 | −0.562 | 0.907 | 101.782 | | |
| Off-route travel in scenic areas can enhance relationships and friendships. | 3.545 | −0.881 | −0.307 | 0.902 | 92.927 | | |
| **Change-locus-of-control argument (CLO)** | | | | | | 0.925 | 0.805 |
| Others still choose to travel off-route in scenic areas, even if I do not do that; | 3.499 | −0.688 | −0.349 | 0.911 | 111.694 | | |
| If I choose to travel without deviating from the route in a scenic area, it makes no difference; | 3.421 | −0.418 | −0.297 | 0.905 | 106.517 | | |
| This scenic environment is also difficult to change when I choose to travel alone in a scenic area without deviating from the route. | 3.486 | −0.439 | −0.332 | 0.876 | 68.981 | | |

Four main first-order variables were proposed based on the review of the literature on the neutralization technique theory [54]. The formative second-order model of neutralization techniques with first-order variables was constructed. The collinearity between formative indicators was also diagnosed using the variance inflation factor (VIF). VIF values for all indicators ranged from 2.014 to 3.072 and were below the recommended value of 5, thereby indicating no multicollinearity. The path coefficients of all variables were significant at the 0.001 level. The analysis results are shown in Table 4. The second-order model was thus built [54].

**Table 4.** Test Results of the Second-order Measurement Model.

| Second-Order Construct | First-Order Constructs | Path Coefficient | t-Value |
|---|---|---|---|
| Neutralization techniques (formative) | Denial of responsibility | 0.310 | 25.378 |
| | Denial of injury | 0.327 | 30.946 |
| | Appeal to higher loyalties | 0.340 | 26.251 |
| | Change-locus-of-control argument | 0.297 | 21.603 |

### 5.2. Testing of Structural Models

The model testing results are shown in Table 5. Behavioral intention is positively affected by attitude, subjective norms, and perceived behavioral control. Path coefficients were β = 0.235, β = 0.231, and β = 0.222 ($p < 0.001$), verifying H1, H2, and H3, respectively. Behavioral intention and perceived behavioral control have a positive impact on uncivilized tourism behavior. Path coefficients were β = 0.242 and β = 0.235 ($p < 0.001$), verifying H4 and H5, respectively. Neutralization techniques have a negative impact on cognitive dissonance. The path coefficient was β = −0.528 ($p < 0.05$), verifying H8. Cognitive dissonance has a negative impact on attitude. The path coefficient was β = −0.381 ($p < 0.001$), verifying H9. Moreover, the negative impact of the cognitive dissonance on uncivilized tourism behavior was also supported by the data. The path coefficient was β = −0.337 ($p < 0.001$), verifying H11.

**Table 5.** Test Results of Path Relationships.

| Hypotheses | Path Coefficient | SE | t-Value | *p* | Hypothesis Testing |
|---|---|---|---|---|---|
| BA→BI | 0.235 | 0.051 | 4.633 | *** | Supported |
| SN→BI | 0.231 | 0.049 | 4.675 | *** | Supported |
| PBC→BI | 0.222 | 0.048 | 4.647 | *** | Supported |
| BI→PH | 0.242 | 0.049 | 4.913 | *** | Supported |
| PBC→PH | 0.235 | 0.043 | 5.481 | *** | Supported |
| NT→CD | −0.528 | 0.050 | 10.602 | *** | Supported |
| CD→BA | −0.381 | 0.052 | 7.345 | *** | Supported |
| CD→PH | −0.337 | 0.045 | 7.429 | *** | Supported |

Note: *** $p < 0.05$.

### 5.3. Testing of Mediating Effect

In the current study, the Bootstrap method was employed to further test the hypotheses on the mediating effect. The specific testing process referred to the study of Hair et al. [54]. Firstly, the mediating effect of behavioral intention was examined. It can be seen from Table 6 that the confidence interval of perceived behavioral control→behavioral intention→uncivilized tourism behavior did not contain 0. Thus, behavioral intention plays a mediating role, supporting H6. Secondly, the confidence interval of subjective norms→cognitive dissonance→uncivilized tourism behavior and that of neutralization techniques→cognitive dissonance→attitude did not contain 0 either. This suggests that cognitive dissonance plays a mediating role in the above paths, thus supporting H7 and H10.

**Table 6.** Test Results for the Mediating Effect.

| Mediating Path | Effect | SE | T | *p* | Bias-Corrected 95% CI | |
|---|---|---|---|---|---|---|
| | | | | | Lower | Upper |
| PBC→BI→PH | 0.054 | 0.015 | 3.672 | *** | 0.033 | 0.092 |
| SN→CD→PH | 0.059 | 0.017 | 3.513 | *** | 0.03 | 0.095 |
| NT→CD→BA | 0.201 | 0.035 | 5.79 | *** | 0.139 | 0.273 |

Note: *** $p < 0.05$.

## 6. Research Conclusions and Significance

### 6.1. Conclusions and Discussion

The effectiveness of the TPB-extended model in explaining the civilized tourism behavior of tourists has been proven in this study. Cognitive dissonance has also been shown to be an important determinant of tourists' civilized tourism behavior. However, few studies have predicted the uncivilized tourism behavior of tourists in a model. The neutralization technique theory has also been applied to the research of behavior-violating norms but is seldom used in the study of uncivilized tourism behavior. Therefore, this study

integrated TPB and the theories of cognitive dissonance and neutralization techniques, as well as contributed to the literature on uncivilized tourism behavior. The latter was achieved, namely, by the planned behavior model based on the perspectives of cognitive dissonance and neutralization. This new perspective broadens the theoretical scope of the research on uncivilized tourism behavior.

First, the research results demonstrate the effectiveness of TPB as a rational framework in analyzing the intention of uncivilized tourism. Attitude has the most significant influence on behavioral intention, followed by subjective norms. Some tourists, i.e., those who approve of uncivilized tourism behavior and feel social pressure more, tend to engage in the behavior [55,56]. Previous studies have confirmed that attitude and subjective norms are predictors of behavioral intention [57,58]. Perceived behavioral control has a significant effect on behavioral intention. Tourists have a higher possibility of conducting uncivilized tourism behavior when believing that they have the opportunity and resources to engage in behavior and are able to control the results as expected [59]. Thus, increasing the resource control of tourists in scenic spots can reduce the generation of behavioral intention. Additionally, behavioral intention plays a mediating role between perceived behavioral control and behavior. This indicates that the behavior of tourists will be transmitted along with the mediating effect of behavioral intention when their control over behavior is stronger, subsequently leading to the generation of more uncivilized tourism behavior.

Second, TPB has fully interpreted the relationship between attitude and behavioral intentions. Nevertheless, existing studies have ignored the predictive effect of behavioral intention on behavior to some extent [4,5,15]. This study confirmed the basic hypothesis of TPB, whereby behavioral intention is highly correlated with behavior. Tourists are very likely to engage in uncivilized tourism behavior once they begin having related behavioral intentions.

Third, little research has examined the relationship among the cognitive dissonance, attitude, and behavior of tourists despite the important role of cognitive dissonance in the formation of attitude [20]. Adding cognitive dissonance to TPB can aid with further understanding the process of cognitive dissonance, as well as the other variables that affect behavior. The results show that cognitive dissonance has a negative effect on attitude and behavior. Tourists can reduce cognitive dissonance by not only taking a positive attitude, but also by engaging in uncivilized behavior, thereby suggesting that cognitive dissonance is a parallel predictor of behavior.

Fourth, subjective norms have an impact on uncivilized tourism behavior via cognitive dissonance. This shows that the higher the subjective norms are, the more behavior will be conducted with the transmission of the mediating effect. When engaging in uncivilized tourism behavior, tourists are influenced by important reference groups, which have expectations of uncivilized tourism behavior. In contrast, certain studies have found that the relationship between subjective norms and behavior has not yet been established due to the influence of individualistic cultures [60].

Fifth, this study contributed to the expansion of TPB. Neutralization techniques and cognitive dissonance were integrated into the TPB model. In addition, the strategies of tourists to reduce dissonance were examined, and the process of defending uncivilized tourism behavior by tourists was answered. Neutralization techniques can effectively reduce cognitive dissonance and influence behavior through attitude. To avoid cognitive dissonance, tourists use neutralization techniques to rationalize their violating social norm behavior. By doing this, they are able to maintain self-identity [15], thus allowing for the continuation of further uncivilized tourism behavior. The research results are in line with the research of Uba and Chatzidakis [22], where neutralization techniques were found to be important factors in predicting the car use behavior of students. This study, in turn, provides valuable insights for the academic community.

*6.2. Management Enlightenment*

Understanding the causes of tourists' uncivilized tourism behavior is of great importance to solve the management dilemma of tourism destinations. Based on the above research, all the variables involved in this study have a significant impact on the uncivilized tourism behavior of tourists. Controlling these variables is beneficial to the sustainable development of tourism destinations.

First, it is necessary to enhance the important roles of attitude, subjective norms, and perceived control in the behavioral intention and uncivilized tourism behavior of tourists. This can be achieved by the following: (1) Establish feedback channels and make timely responses. Allow tourists to raise suggestions about the problems that exist in facilities, services, and information resources (e.g., satisfaction with services and the lack of public facilities). Satisfy the catharsis of tourists' bad emotions, as well as help tourists to realize the importance of their opinions to improve their attitude toward tourism destinations. (2) Introduce the content of and the harm found in violating norms and local customs to tourists by providing tourism information systems (e.g., tour guide explanations, publicity brochures, billboards), as well as develop other ways in which to make tourists realize the serious consequences of their uncivilized tourism behavior. (3) Guide and transform the uncivilized tourism behavior of tourists, set the best areas for photography, graffiti, etc., and remind tourists to take pictures and leave messages in correct locations in order to reduce free climbing, picture taking, and other such behaviors. (4) Encourage tourists to recycle rubbish spontaneously and help them to develop good habits through the incentive system (by reducing or remitting admission fees, or by providing fees for tourism items).

Second, tourists take advantage of neutralization techniques to justify their uncivilized tourism behavior. Certain interventions can reduce the use of neutralization techniques: (1) For tourists using appeal to higher loyalties and change locus of control arguments, those tourists engaging in civilized tourism should be praised or rewarded, which may change their evaluation of relating to others and social normality. (2) To avoid criticism from tourists, tourism destinations should increase their infrastructure and service level. Special service centers can be set up to provide tourists with free consultation, maps, brochures, and other types of information. (3) For tourists who claim that bad behavior is innocuous, tourism destinations and tour guides should play a guiding and supervising role. The uncivilized tourism behavior of tourists can be restrained by norms, thus making tourists realize their responsibilities.

Third, cognitive dissonance is a predictor of uncivilized tourism behavior. The dissonance of tourists can be reduced in the following specific ways: (1) design anthropomorphic images, use intelligent technology to integrate them into reality, increase the participation of tourists in interactive activities, and enable tourists to intuitively understand the adverse impact of their behavior; and (2) provide humanized safety and civilization tips, use friendly advocacy slogans, and make it easy for tourists to accept, psychologically, reducing the occurrence of uncivilized tourism behavior.

## 7. Limitations and Prospects

Certain limitations still exist even though the research conclusion is conducive to understanding the uncivilized tourism behavior of tourists. Firstly, a certain degree of deviation from social expectations may exist in tourists' self-reports of sensitive information, i.e., the respondents may have concealed their true thoughts in order to meet social expectations (identity, status, occupation, etc.). It was found that the uncivilized tourism behavior of college students occupies a high proportion of the sample, thereby indicating that this group will actively remove or transfer sensitive information. Secondly, the cross-sectional data in this study hindered the examination of causality. Future studies can make inferences about the temporal dynamics of tourist tourism. For instance, research can be conducted on whether the use of neutralization techniques precedes or follows uncivilized tourism behaviors. In addition, the causal relationships between constructs can be explored through longitudinal data analysis. Finally, Goh et al. [4] studied the uncivilized tourism behavior

of tourists and found that individual misbehavior is related to activity places. Whether other factors trigger or deter uncivilized tourist behavior of tourists remains unclear [1]. As such, future research should further explore the mechanisms that drive this behavior.

**Author Contributions:** Conceptualization, K.C. and P.Z.; methodology, P.Z.; software, P.Z.; writing—original draft preparation, P.Z.; writing—review and editing, P.Z. and K.C.; funding acquisition, K.C. All authors have read and agreed to the published version of the manuscript.

**Funding:** This research was co-funded by the National Natural Science Foundation of China, No. 42161036; the Xinjiang Uygur Autonomous Region Postgraduate Research Innovation Project, No. XJ2022G057.

**Institutional Review Board Statement:** Not applicable.

**Informed Consent Statement:** Not applicable.

**Data Availability Statement:** The data presented in this study are available on request from the corresponding author.

**Acknowledgments:** We are also grateful to the editors and reviewers for their helpful comments and support of the publication.

**Conflicts of Interest:** The authors declare no conflict of interest.

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
