# Peer review of "Understanding the Uncivilized Tourism Behavior of Tourists: A Planned Behavior Model Based on the Perspectives of Cognitive Dissonance and Neutralization"

_sustainability, doi:10.3390/su15054691_

Round 1
Reviewer 1 Report
REVIEWER REPORT
Reviewer Report for the Manuscript: Sustainability- 2211854
“Understanding of Tourists’ Uncivil Behavior: A Model of Planned Behavior Based on a Cognitive Dissonance and Neutralization Perspective”
Journal Name: Journal of Sustainability
Summary
The article aims to investigate a very important in sustainable tourism management. It analysis the uncivilized behavior of tourists using three main basic theories and inferential statistical analyses. The authors investigate the factors affecting the uncivilized behavior of tourists. The work is valuable and the results can contribute to the literature. However, there are some concerns regarding different parts of the article.
Abstract
1. This section lacks an overall method of research. Giving the highlights of the article methodology is important for better comprehension of readers.
2. The sentence “This paper provides a better understanding of this phenomenon in academia and industry” is somehow incomplete. Please either complete or delete
Introduction
3. This section should be started with uncivilized tourism behaviors discussion and concern and be focused on the subject by reviewing the history of the concern in tourism destination management literature.
4. There is an ambiguity seen in this section as it reviews some literature while there is the other section for this purpose. Therefore, this section is better to focus on a compelling argument on the root causes and effects of uncivilized tourism behaviors and TPB.
5. This section lacks contributions of the study to concurrent literature in both theory and methodology sides.
Literature Review
6. Three main theories of the research have been reviewed at the end of this section which is good. However, their main benefits for using in the article should be summarized in a table.
Research Hypothesis and Model Construction
7. I strongly suggest that authors combine hypotheses since the high number of them would lead to some confusing results. Moreover, due to nearing most of the concepts and relationships brought in the hypothesis, they can be reduced to three or four without damaging the analysis and results.
8. The hypothesis “H3: Cognitive dissonance has a negative impact on the subjective norms of tourists’ uncivil behavior” is too clear to be a hypothesis as there always is a direct relationship between Cognitive dissonance and subjective norms.
Sample Description
9. There should be a justification for the majority of men in 387 valid samples as gender equality in the sampling, investigating and analysis are areas of concern.
Presentation
10. There should be a map presenting the location of the study area.
Management Insights
11. The results can be discussed in the line with sustainable management of tourism destinations giving some suggestions for improvement in the form of policy implications.
Reviewer 2 Report
The study is interesting, presenting an innovative approach to a problem that needs investigation and being pleasant to read. However, I think that it could still improve, above all, in its theoretical implications and reveals, in my opinion, some lack of foundation in the options taken in the construction of the structural model. So, I present my comments below:
page 1-2 - Introduction: I think it would be important to better explain what uncivilized tourism behaviours are, if possible, referring to examples.
page 4 - line 162 - It would be important to exemplify neutralization techniques.
page 6 - line 239 - Who are the tourism experts? Authors should introduce them.
page 6 - line 244 - It would be important to justify the reason for choosing 4 dimensions associated with tourist neutralization techniques.
Page 11 - Regarding the structural model, I have some doubts... Are there no correlations between the dimensions in the final structural model? Are there no mediation effects in the model? To obtain a better fit in the final model, shouldn't some trajectories be eliminated? It is important to explain these aspects.
page 12-13 - This article presents a discussion of the results that is well thought out and contrasted with the literature. However, it does not clearly show the contribution that the study brings to the cinematic community. What advances does this study bring to the state of the art?
Round 2
Reviewer 1 Report
The revised article is fine now. However, the references are mostly older than 2015 which needs to be updated.
Reviewer 2 Report
Significant improvements have been made to this article. Thus, in my opinion, this article already meets the conditions to be publishable.
